# Hematopoiesis, Inflammation and Aging—The Biological Background and Clinical Impact of Anemia and Increased C-Reactive Protein Levels on Elderly Individuals

**DOI:** 10.3390/jcm11030706

**Published:** 2022-01-28

**Authors:** Øystein Bruserud, Anh Khoi Vo, Håkon Rekvam

**Affiliations:** 1Department of Clinical Science, University of Bergen, 5020 Bergen, Norway; hakon.reikvam@uib.no; 2Department of Medicine, Haukeland University Hospital, 5021 Bergen, Norway; khoiavo93@gmail.com

**Keywords:** anemia, hematopoiesis, inflammation, aging, C-reactive protein, survival

## Abstract

Anemia and systemic signs of inflammation are common in elderly individuals and are associated with decreased survival. The common biological context for these two states is then the hallmarks of aging, i.e., genomic instability, telomere shortening, epigenetic alterations, loss of proteostasis, deregulated nutrient sensing, mitochondrial dysfunction, cellular senescence, stem cell exhaustion and altered intercellular communication. Such aging-associated alterations of hematopoietic stem cells are probably caused by complex mechanisms and depend on both the aging of hematopoietic (stem) cells and on the supporting stromal cells. The function of inflammatory or immunocompetent cells is also altered by aging. The intracellular signaling initiated by soluble proinflammatory mediators (e.g., IL1, IL6 and TNFα) is altered during aging and contributes to the development of both the inhibition of erythropoiesis with anemia as well as to the development of the acute-phase reaction as a systemic sign of inflammation with increased CRP levels. Both anemia and increased CRP levels are associated with decreased overall survival and increased cardiovascular mortality. The handling of elderly patients with inflammation and/or anemia should in our opinion be individualized; all of them should have a limited evaluation with regard to the cause of the abnormalities, but the extent of additional and especially invasive diagnostic evaluation should be based on an overall clinical evaluation and the possible therapeutic consequences.

## 1. Introduction

The ageing global population is regarded as the most important present and future medical and social demographic problem worldwide by the World Health Organization [1]. Recent estimates suggest that 38% of the word population will be aged at least 65 years in 2050, and it is also estimated that in 2050 there will be a larger number of older people aged at least 60 years than adolescents aged 10–24 years (2.1 billion versus 2.0 billion). Thus, the optimal handling of medical problems in the aging population is already now a challenge and will become even more challenging during the next decades.

The complex process of aging is characterized by the modulation of fundamental cellular processes, and this is reflected in the previously described nine hallmarks of aging, which include genomic instability, telomere attrition, epigenetic alterations, loss of proteostasis, deregulated nutrient sensing, mitochondrial dysfunction, cellular senescence, stem cell exhaustion and altered intercellular communication (Figure 1) [2,3,4,5,6,7,8,9,10,11]. These cellular effects lead to aging-dependent alterations in organs and tissues, including hematopoietic cells together with their supporting stromal cells in the common bone marrow microenvironment, as well as various immunocompetent cell subsets with the modulation of their immunoregulatory interactions [2,12,13,14,15,16]. Aging can thus alter the regulation of both hematopoiesis and inflammation [12]. In this context we describe and discuss the effects of aging on normal hematopoiesis together with the occurrence of anemia and increased C-reactive protein (CRP) levels in elderly individuals. We would expect the frequency and causes of anemia to differ between developed and underdeveloped countries; we therefore emphasize that the present review is mainly based on studies in developed countries.

## 2. The Biological Context of Anemia in Elderly Individuals: Hallmarks of Aging in Normal Hematopoietic Cells and Their Associations with Signs of Inflammation

The mechanisms involved in hematopoietic stem cell aging have previously been classified as stem-cell-intrinsic (i.e., alterations in the hematopoietic cells) and stem-cell-extrinsic (i.e., indirect effects mediated by aging bone marrow stromal cells) [12,14,15,16]. However, the various mechanisms are interconnected, and it is therefore difficult to maintain this strict classification [12]. The present article gives a relatively brief overview of the important mechanisms involved in aging hematopoiesis; for more detailed discussions and additional references we refer to several recent excellent reviews [12,13,14,15,16,17].

### 2.1. The Bone Marrow Microarchitecture and the Stem Cell Pool in Aging

The hematopoietic stem cell distribution in the bone marrow is altered during aging; the number of stem cell niches and hematopoietic stem cells decreases close to the bone surface (i.e., the endosteum), but they expand further away from the bone compared with younger individuals [18]. The stem cells/niches are also more distant from arterioles and megakaryocytes in aging, whereas perisinusoidal niches seem to be preserved and have a distance from sinusoids similar to younger individuals [19,20,21]. This altered microarchitecture is at least partly caused by decreased noradrenergic innervation in the bone marrow, where β_2_-adrenergic–IL6-dependent megakaryopoiesis is important for the close localization of stem cells to megakaryocytes [19,20]. Stem cell lodging to certain non-endosteal niches thereby seems to be favored.

The number of phenotypic hematopoietic stem cells increases upon aging, but their regenerative potential decreases and they preferentially differentiate into myeloid cells and less into lymphoid cells [12,20,22,23,24,25,26]. The β_2_/IL6 axis is also important for the regulation of the more differentiated myelopoiesis, and in experimental studies adrenergic stimulation can decrease this myeloid dominance [19,20]. Geiger and van Zand [25] suggested two decades ago that aging mainly affected hematopoietic quality rather than its quantity. Their statement was based on the observation that the hematopoietic stem cell population is increased during aging and able to maintain normal peripheral blood cell counts throughout life, but seems to lack the “functional reserve” needed during crises [25,27,28]. This may also (at least partly) explain why aging hematopoiesis with the development of anemia can be a strong comorbidity factor for several other diseases [22].

### 2.2. Hematopoiesis and Hematopoietic Stem Cells in Elderly Individuals: Cell-Intrinsic Mechanisms Involved in Stem Cell Aging

Several cell-intrinsic mechanisms are involved in the aging of hematopoietic cells. Table 1 presents a summarizing overview of important intrinsic mechanisms that are important in the aging of hematopoietic cells. Several of these mechanisms will also influence the regulation of inflammation and thereby contribute to the regulation of both hematopoiesis and inflammation. A more detailed discussion and additional references are included in Section 2.2.

#### 2.2.1. Genetic Instability, Telomere Shortening, Altered Cell Communication and Inflammation

Random DNA damage accumulates in hematopoietic stem cells as a part of the aging process [3,12]; this should be regarded as a sign of genomic instability, which is one of the hallmarks of aging (Figure 1) [2,3]. Experimental studies suggest that accumulating DNA damage is an important mechanism in hematopoietic stem cell aging, and animal models suggest that altered functions of the mechanisms involved in genomic maintenance are important for this accumulation, e.g., nucleotide excision repair, telomere maintenance and non-homologous end-joining [27]. DNA damage then seems to accumulate in stem cells with age [27]. These abnormalities/deficiencies do not seem to deplete the stem cell reserve; they rather manifest as functional stem cell deficiencies under conditions of stress, e.g., wound healing, hematopoietic ablation [27,28].

DNA damage leading to telomere shortening is possibly a specific hallmark of aging that leads to stem cell aging/exhaustion/abnormalities [4], but telomere defects alone cannot explain all the signs of stem cell aging (Figure 1) [2,28]. Additional specific mutations also contribute to clonal expansion and the emergence of clonal hematopoiesis [14], an age-associated abnormality that can possibly develop later into hematological malignancies [29]. This hypothesis is supported by the observation that myelodysplastic syndromes (MDS) as well as pre-MDS stem cells show a higher degree of subclonal complexity than normal cells, including aging-associated variants [29]. However, it is controversial whether mutations associated with clonal hematopoiesis are truly oncogenic or whether they only increase self-renewal and thereby make it more likely for additional and truly oncogenic mutations to occur later in these actively self-renewing stem cells.

Patients with clonal hematopoiesis have an increased risk of atherosclerosis and cardiovascular complications [12,30,31]. This observation is consistent with the hypothesis that fully differentiated cells (especially monocytes/macrophages) in these patients have abnormal functions and thereby predispose them to inflammation with increased CRP levels and progression to atherosclerosis with clinical manifestations [32,33]. The hypothesis is supported by observations in patients with *TET2* mutations who show increased inflammasome-mediated IL1 secretion by monocytes and thereby a predisposition to inflammation and atherosclerosis [30]. Inflammation induced by clonal hematopoiesis and immune cell dysfunction thus seems to contribute to the association between CRP increase (i.e., acute-phase reaction) and cardiovascular disease [33]. However, an alternative explanation could be that mutations and clonal hematopoiesis occur more frequently in myeloid cells exposed to a chronic proinflammatory microenvironment. Whether age-dependent alterations in erythrocytes (e.g., abnormalities similar to storage lesions [34]) and/or platelets contribute to this predisposition to cardiovascular complications in patients with clonal hematopoiesis is not known. Thus, according to these observations genetic instability and altered cytokine-mediated intercellular communication (i.e., two hallmarks of aging, see Figure 1) may be involved in the association between clonal hematopoiesis and cardiovascular complications.

#### 2.2.2. Epigenetic Abnormalities, Epigenetic Drift and Inflammation

The epigenome and the organization of chromatin differ between younger and aged hematopoietic stem cells [12,14], and mutations associated with clonal hematopoiesis are often seen in genes involved in epigenetic regulation [14]. These epigenetic changes have been discussed in detail previously [14,23,24]; they include both posttranscriptional histone modulation and chromatin organization as well as DNA methylation with site- or gene-specific modulations, e.g., the hypermethylation of genes regulated by polycomb repressive complex 2 [23,35,36]. Previous studies have described an overall increase in transcriptional activity that is at least partly caused by altered epigenetic regulation; these authors described the increased expression of genes associated with stress responses, inflammation and protein aggregation, whereas genes involved in the preservation of genomic integrity and chromatin modeling showed reduced expression [24]. Finally, age-dependent histone modifications seem to alter the expression of genes involved in the regulation of the proliferation, self-renewal, differentiation and maintenance of hematopoietic stem cells [36].

Epigenetic drift has been defined as all changes with a general effect on the epigenome and chromatin organization/architecture [37,38]; it seems to be a part of the aging process and to occur across tissues [38], and would be expected to include hematopoietic cells (Figure 1) [5,23,24,38]. Studies in animal models suggest that epigenetic drift is associated with the development of inflammation [38], and this is further supported by studies on aged human mononuclear cells [39]. Studies on aging immunocompetent cells show specific epigenetic signatures in different immunocompetent cells (T and B lymphocytes, NK cells and monocytes), and variations in signatures between individuals also seem to increase with age [39]. These histone/chromatin variations between individuals as well as cell-to-cell variations can be detected in stem, progenitor and differentiated cells, and it has been suggested that variations between mature cells arise from variations between distinct hematopoietic stem cell clones [39]. It has also been suggested that such epigenetic variations in immunocompetent cells (together with the altered balance between various lymphocyte subsets) contribute to the development of inflammaging and/or an increased risk of severe infections with increased morbidity/mortality in elderly individuals [37].

#### 2.2.3. Cellular Polarity and Epigenetic Asymmetry in Hematopoietic Stem Cell Division

Several molecules appear to be polar in the cytoplasm and nucleus of young hematopoietic stem cells, and high levels of the two cytoplasmic molecules cell division control protein 42 (cdc42) and laminin, together with the nuclear polarity of epigenetic markers, are regarded as intrinsic markers of altered polarity and stem cell aging [18,40,41]. This polarity of the cells allows hematopoietic stem cells to undergo asymmetric cell division, i.e., one daughter cell that differentiates and another daughter cell that retains the stem cell potential [41]. This polarity seems to be reduced or lost as a part of the aging process, and this is true both for cytoskeletal and cell cycle regulatory proteins in the cytoplasm as well as epigenetic markers in the nucleus [14,18,40,41]. Hematopoietic stem cells maintain a balance between self-renewal and differentiation, and the premitotic polarity status seems to be important for this balance in addition to the outcome after hematopoietic stem cell division [41]. Aged apolar hematopoietic stem cells preferentially go through symmetric divisions, resulting in daughter cells with reduced regenerative and lymphoid potentials, whereas younger polar cells preferentially undergo asymmetric division and thereby maintain cells with stem cell potential [41].

#### 2.2.4. Metabolic Characteristics and Regulation of Protein Homeostasis

During their development hematopoietic cells go through multiple differentiation steps and transition through several microanatomical sites that require metabolic shifts [42]. Even though quiescent and cycling hematopoietic stem cells show similar high glycolytic activities, they differ in several metabolic characteristics, e.g., quiescent cells show higher lysosomal activity and autophagy/mitophagy whereas cycling stem cells show higher protein synthesis, ATP production and intracellular levels of reactive oxygen species [42]. The metabolic balance is altered in aging hematopoietic stem cells, which show a general shift to a higher rate of oxidative metabolism [16,42]. The aged cells also show altered protein homeostasis; this process is regulated by several cellular mechanisms, including autophagy (i.e., a cellular response to stress, for example, metabolic adaptation) and the ubiquitin/proteasome system, and both these systems are reduced in aged cells (Figure 1) [7,43,44,45]. Aging is thereby associated with the cellular accumulation of misfolded or damaged proteins [43] because the endoplasmic reticulum stress response (also called the unfolded protein response) is reduced [44]. Finally, aging-dependent alterations in sex hormone levels seem to represent an additional systemic mechanism for the downregulation of this stress response [44].

#### 2.2.5. Senescence and Intracellular Signaling

Cellular senescence is regarded as a stress-induced irreversible growth arrest, and it is often characterized by a distinct secretory profile, i.e., an altered communication with the neighboring cells [46]. Irreversibly cell-cycle-arrested senescent cells accumulate during normal aging, and animal models suggest that these cells actively participate in the development of aging-associated organ deterioration and may further increase the aging-dependent risk of malignant diseases (Figure 1); in these animal models the elimination of senescent cells therefore increases the life span [46]. This effect was also seen for normal hematopoietic cells, where the pharmacological elimination of senescent hematopoietic stem cells counteracted the aging-dependent reduction in the regenerative potential of hematopoietic stem cells [46,47]. Furthermore, signaling through several intracellular pathways seems to be altered during aging due to intrinsic mechanisms, e.g., TGF1β, Notch, NFκB and Wnt signaling [24,48,49,50,51]. Both the induction of senescence with an altered secretory profile and the altered intracellular signaling downstream to cell surface receptors may represent combined direct and indirect effects on hematopoiesis [12,14], including the effect of increased senescence on aging with altered mediator secretion and thereby the modulation of autocrine/paracrine circuits (Figure 1) [2,9].

### 2.3. Stem-Cell-Extrinsic Mechanisms Involved in Hematopoietic Aging: Stem Cell Niches, Stromal Cell Subsets and Cellular Communications

Normal hematopoietic cells have a hierarchical organization and are supported by various non-hematopoietic stromal cells that also form stem cell niches where the minor population of hematopoietic stem cells are maintained [16,17,20]. The most important stromal cells that contribute to the stem cell niches are:Mesenchymal stem cells (MSCs). These cells are located close to arterioles and more loosely around sinusoidal vessels [17]; they are heterogeneous, and two main populations have been identified based on their expression of platelet-derived growth factor receptor (PDGFR)α and stem cell antigen 1 [17]. The MSCs support normal hematopoiesis through several mechanisms that are modulated by MSC aging, including their supportive function in stem cell niches (Figure 2, Table 2) [16,19,52,53,54,55,56,57,58,59,60,61]. Several epigenetic mechanisms are important for MSC aging and the alterations of their hematopoiesis-supporting mechanisms, including both altered DNA methylation and histone modification (e.g., acetylation) [61]; increased senescence is also observed [19,52,53,54,55,56,57,58,59,60,61]. First, MSCs produce several soluble mediators that are important both for myelopoiesis (e.g., CXCL12) and lymphopoiesis (e.g., IL7) [52]; the release of several growth factors is thus reduced. Second, MSCs and sinusoidal endothelial cells seem to form a complex network in close contact with the extracellular matrix and pervading the marrow tissue. The structural features of stromal components are maintained during aging [53], and although central perisinusoidal MSCs are increased or maintained there is a reduction in periarteriolar MSCs [16,19,56]. Third, the MSC functions in these networks seem to be altered, especially with regard to the regulation of cell cycle progression of the stem cells, stem cell trafficking in the microenvironment and the localization of progenitors close to different MSC subsets with different perivascular localizations [57]. Finally, the adipogenic differentiation of MSCs is preferred, and another consequence of this aging effect is reduced bone formation [59].Osteoblasts and other osteolineage cells. Osteoblasts are the predominant bone-lining endosteal cells [54], whereas osteolineage or osteoblastic lineage cells refer to the intermediate stages of differentiation in the direction from MSCs towards osteoblasts [57]. These cells and particularly mature osteoblasts seem to be the most important for the maintenance of more committed progenitors, especially lymphoid cells [62,63]. They stimulate/regulate hematopoiesis both through cell–cell contact (e.g., expression of the Notch ligand Jagged1) and through the release of soluble mediators (e.g., the growth factors CXCL12, stem cell factor and angiopoietin 1; osteopontin) [17]. Aging causes a decrease in the number of osteoblasts and in addition decreases osteopontin release via these aging cells (Figure 2, Table 2) [19,55,62,63]. As described above, aging MSCs favor adipogenic differentiation [59], and a reduction in osteoblasts is then caused by several mechanisms, including the induction of apoptosis, the increased release of reactive oxygen species, decreased glutathione reductase activity and the increased phosphorylation of p53 and p66 [64]. Finally, animal models suggest that a reduction in/lack of osteopontin causes decreased engraftment capacity but increases long-term stem cell frequency together with loss of stem cell polarity; as would then be expected from these observations, thrombin-activated osteopontin attenuates the aging stem cell phenotype [55].Adipocytes. Adipocytes release factors that seem to inhibit hematopoiesis [65,66,67]. Aging accelerates bone marrow adipogenesis [53]. This is apparent especially in the long bones where hematopoietic marrow is gradually replaced by adipocyte-rich marrow; although adipocytes release certain supportive mediators their overall effect is a reduction in hematopoiesis (Figure 2, Table 2) [19,66,68,69]. Animal studies suggest that this aging-associated adipocyte expansion can be further increased by dietary fat intake in aged animals [69]. The process of favored differentiation into adipocytes seems to be regulated at the transcriptional level and involves the transcriptional regulators Maf and Runx2 [58]. Furthermore, the release of adiponectin is a possible mechanism for the inhibition of hematopoiesis by adipocytes because this mediator has an antiproliferative and possibly also a proapoptotic effect, especially on myelomonocytic lineage cells [65]. The pharmacological inhibition of adipogenesis has therefore been suggested as a possible strategy to reduce the negative effects of adipogenesis on normal hematopoiesis [66,67]. However, the inhibitory effect of adipocytes may depend on the biological context, as a recent animal study has shown that adipocytes or a subset of adipocytes could release SCF and thereby promote regeneration after irradiation and myelotoxic chemotherapy [68].Endothelial cells. Endothelial cells and perivascular cells are intimately connected. Arteriolar and sinusoid endothelial cells seem to differ in their mechanisms with regard to supporting hematopoiesis (Figure 2) [54]; in particular, arteriolar cells release a wide range of hematopoietic growth factors [17,70,71]. The aging of endothelial cells has multiple effects on the stem cell niche and normal hematopoiesis (Table 2) [19,53,54,55,56,57,72,73]. First, the bone marrow endothelium shows aging-associated morphological and metabolic changes, including increased levels of reactive oxygen species that decrease their angiogenic and migratory potential, and the microvessels show a loss of integrity with augmented leakiness [70,71]. Second, the decreased release of prohematopoietic soluble mediators, including SCF and CXCL12, is one of the endothelial contributions to hematopoietic aging [70,72,74]. Third, the reduction in niche-forming vessels is likely to induce metabolic changes in the bone marrow microenvironment [70]. Finally, the niche-forming vessels in aging mice can be restored either by endothelial transplantation [72] or by the activation of endothelial Notch signaling, which seems to be altered in the aging bone marrow endothelium [56].Perivascular cells. This cell population is heterogeneous and includes cells expressing both pericyte and smooth muscle markers [17,70]. Aging reduces the abundance of pericytes and thereby the release of several soluble mediators that are important for the induction of quiescence of hematopoietic stem cells (e.g., SCF, bone morphogenic proteins 4 and 6) [70].

Neural regulation. Sympathetic and sensory nerves innervate both the bone and the bone marrow [27]. Furthermore, human CD34^+^ cells express both dopaminergic and β2 adrenergic receptors; the receptors are expressed especially by immature CD34^+^CD38^low^ cells and can be upregulated by G-CSF and GM-CSF [74]. Thus, adrenergic signals act directly on human hematopoietic progenitors and can increase their migration, proliferation, polarity as well as extracellular protease release, and Wnt-initiated signaling is involved in this stem cell modulation [74]. The perivascular arteriolar niche consists of specialized MSCs together with adrenergic nerves and megakaryocytes, and these cells are closely associated with quiescent stem cells [62,75,76,77]. Finally, nonmyelinating Schwann cells (i.e., glial fibrillary acidic protein-expressing cells) ensheath autonomic nerves, express genes that are important for the support of hematopoietic stem cells and can activate the latent form of TGFβ [76]. Thus, autonomic nerves are not only important through the direct effects of neurotransmitters on hematopoietic cells but also through their modulation of the niche cytokine network [76] and indirectly through the modulation of adrenoreceptor-expressing MSCs [77]. There is an aging-associated sympathetic denervation of the niche, and targeting this denervation with adrenoreceptor β3 agonists improves the function of aged stem cells in animal models [19,20,78]. Another effect of the denervation is the expansion of MSCs with decreased stem cell supporting capacity, a reduction in arterioles and increased stem cell numbers [16,19,54,57,74].Megakaryocytes. The role of megakaryocytes in the regulation of normal hematopoiesis can be regarded as a feedback mechanism. Megakaryocyte precursors migrate from the endosteal microenvironment to sinusoids for maturation, and noradrenergic bone marrow innervation promotes β2-adrenergic/IL6-dependent megakaryopoiesis [20]. A subset of hematopoietic stem cells is then associated with megakaryocytes that regulate stem cell quiescence through the release of soluble mediators (especially CXCL4 and TGFβ) as well as CD41 expression (Figure 2, Table 2) [18,19,37,54,57,79,80,81,82,83]. Age-dependent epigenetic alterations in hematopoietic stem cells and possibly also megakaryocytes seem to modify these interactions between megakaryocytes and neighboring hematopoietic stem cells [37]. Thus, there seems to be an interaction between aging, sympathetic innervation/denervation, epigenetic modulation and megakaryopoiesis with regard to the effects of aging on hematopoiesis [37]. The effects of megakaryocyte are partly mediated through the local release of TGFβ, which is important for the regulation of quiescence and initiates SMAD signaling in stem cells [79]; additionally, megakaryocytes release thrombopoietin [80] and CXCR4 [81], which act directly on immature hematopoietic cells. The release of thrombopoietin and possibly also other mediators can be stimulated by the ligation of C-type lectin-like receptor 2 (CLEC-2), and megakaryocyte expression of this receptor thereby becomes important for the regulation of stem cell quiescence [83]. Finally, the peripheral blood platelet count will possibly modulate these megakaryocyte effects through its effects on systemic thrombopoietin levels [37].Neutrophils. Neutrophils also seem to have regulatory functions in normal hematopoiesis; the mechanisms involve the neutrophil-mediated augmentation of sympathetic nervous system effects with the release of prostaglandin E2 [63]. These observations show that neutrophils can function as a link between the sympathetic nervous system and the stem cell niches.Monocytes, macrophages and osteoclasts. Both monocytes and other immunocompetent cells can contribute to the aging of hematopoiesis [17], possibly through their modulation of local levels of various proinflammatory cytokines [60]. Furthermore, aging seems to be associated with a shift from the anti-inflammatory M2 phenotype to the proinflammatory M1 phenotype; this shift is associated with the increased release of proinflammatory cytokines and seems to depend on monocyte/macrophage expression of the Foxo3 transcription factor (i.e., it is probably caused by an intrinsic mechanism) [11]. This shift is associated with local inflammation in the gastrointestinal tract, and in our opinion one should further investigate whether this shift is also important for inflammaging and/or the regulation of aging normal hematopoiesis. Finally, bone-marrow-associated macrophages are also important to maintain many stem-cell-supporting characteristics of MSCs, including their release of CXCL12 and SCF [57]. Finally, osteoclasts support hematopoiesis and lymphopoiesis indirectly by increasing the osteoblast secretion of CXCL12 and IL7 [57].T cells. As described in a recent review, activated T cells release several cytokines involved in the regulation of normal hematopoiesis [63]. CD4^+^ T cells thereby stimulate hematopoiesis, whereas CD4^+^CD25^+^ regulatory T cells seem to inhibit it. Furthermore, the clinical experience from allogeneic stem cell transplantation suggests that T cells facilitate engraftment, and even regulatory T cells (including CD150^+^ Treg cells) seem to facilitate engraftment and promote stem cell quiescence [63]. Finally, animal models suggest that CD8^+^ T cells also contribute to the regulation of hematopoiesis because IL6 and IFNγ released by CD8^+^ T cells can trigger emergency myelopoiesis [63].

These descriptions of the various bone marrow stromal cells and their contributions to the stem cell niches are far from complete, but they clearly illustrate that many different stromal cells form an extensive and complex interacting network through the release of soluble mediators, cell–cell contact and cell–extracellular matrix contact. This hematopoiesis-supporting network is altered during aging. The age-dependent modulation of one stromal component can alter the functions of other stromal cells and will thereby have both direct and indirect effects on hematopoiesis. Furthermore, several soluble mediators released by various stromal cells are important for the aging of hematopoietic stem cells, e.g., CCL5, which shows high levels in the aging stem cell milieu and is involved in the myeloid lineage skewing [84], osteopontin that can induce a loss of cell polarity and reduced engraftment potential [55] and the T-cell- and monocyte-derived cytokines IL1α, IL1β, IL3 and IFNγ, which influence the migration and maturation of megakaryocytes (Figure 2, Table 2) [85].

### 2.4. Myeloid Skewing: An Intrinsic or Extrinsic Effect?

The overall effects of hematopoietic–stromal interactions in aging are illustrated by previous experimental animal studies. Transplanted young hematopoietic stem cells engraft at a lower efficiency when transplanted to aged compared with young recipients [15,84]. Furthermore, coculture experiments show that aged endothelial cells impair the function and increase the myeloid bias of younger hematopoietic stem cells, whereas endothelial cells restored the repopulating capacity of aged hematopoietic stem cells but did not alter the myeloid bias, which seems to be an intrinsic characteristic of stem cells [72]. However, other studies have shown that the IL1-mediated inflammatory (aging-associated) effects on hematopoiesis and hematopoietic stem cells are reversible [85,86,87,88]; the same is possibly true for lymphoid hematopoietic stem cells, which seem to retain their normal lymphoid potential if they are removed from the aging microenvironment that causes the myeloid skewing of hematopoiesis [89]. Thus, hematopoietic aging with myeloid skewing depends both on the aging of the hematopoietic cells themselves and on the aging of the supporting stromal cells.

### 2.5. Inflammation and Hematopoiesis in Aging: The Contributions of Individual Cytokines and A Focus on the Myeloid Skewing

The aging of the bone marrow microenvironment is associated with increased levels of several proinflammatory cytokines, including IL1β, IL6 and TNFα, which are also known as drivers of the acute-phase reaction together with other members of the IL6 family and various chemokines [60,86]. Experimental studies suggest that several age-associated characteristics of normal hematopoiesis are associated with increased proinflammatory cytokine activity caused by increased release by various stromal cells:IL1. IL1α/β exposure can induce myeloid skewing of normal hematopoietic stem cells at the expense of self-renewal [87]. Aging macrophages seem to stimulate megakaryocytic differentiation and myeloid skewing through IL1β-induced signaling [60,88]. IL1 also blocks the lymphoid differentiation of stem cells [89].IL6. This cytokine seems to promote thrombopoiesis (i.e., megakaryocyte modulation) [60].TNFα. This cytokine seems to stimulate myelopoiesis in aging [90].CCL5. This proinflammatory chemokine increases with age and seems to stimulate myeloid-biased differentiation [84].TGFβ and IFNγ. These two cytokines are also regarded as proinflammatory and contribute to megakaryocyte modulation [60,91].A certain subset of hematopoietic stem cells seems to respond to proinflammatory stimuli and thereby becomes particularly important for the myeloid skewing of hematopoiesis [90]. This study also suggests that young and aged long-term hematopoietic stem cells respond differently to inflammatory stress, such that the aged cells show a myeloid-biased gene expression initiated by several transcription factors, including Klf5, Ikzf1 and Stat3 [90].

These observations, together with the increased levels of proinflammatory cytokines in many elderly individuals (see Section 6), strongly suggest that there is an association between the induction of an acute-phase reaction and the development of anemia in aging.

Aging-dependent alterations in normal hematopoiesis are caused by the overall effect of a wide range of factors both in hematopoietic cells and in their supporting stromal cells. There is a complex crosstalk between hematopoietic and stromal cells as well as between various stromal cell subsets; this communication is altered in aging. Our present description of the effects of aging on hematopoiesis is definitely not complete, and for more detailed discussion and additional references we refer to recent excellent reviews [12,13,14,15,16,17,18,19,20,60]. However, our review shows that aging of hematopoiesis is a multifactorial process involving both immunocompetent cells and the regulation of inflammation.

### 2.6. Aging and Leukemic Hematopoiesis: Acute Myeloid Leukemia as an Example

Hematological malignancies are most common in elderly individuals, e.g., acute myeloid leukemia (AML) has a median age at the time of first diagnosis of 65–70 years [92,93]. As outlined above (Section 2.2), clonal hematopoiesis can be detected in elderly individuals and can be regarded as a part of the aging process. The biological characteristics of hematological malignancies seem to be determined not only by cancer-associated genetic abnormalities alone but also by the biological characteristics of the aging process that are transferred from normal to leukemic hematopoietic cells, and the experience with AML suggests that aging is associated with chemoresistance. First, favorable genetic abnormalities are less common in elderly individuals [92,93]. Second, a larger subset of elderly patients has high-risk secondary AML following previous chemotherapy or hematological disease (i.e., MDS, chronic myeloproliferative neoplasia) [92,93]. Third, the biology of AML cells from elderly individuals seems to differ from that of AML cells in younger patients even when the cells have similar AML-associated genetic abnormalities [94]. Thus, aging not only influences the risk but also the biological characteristics of AML; the same may also be true for other malignancies.

## 3. Anemia in Elderly Individuals

### 3.1. Definition of Anemi

The level of hemoglobin varies considerably between healthy individuals and depends on age as well as gender; despite these variations, the level in each individual is relatively stable [95]. The World Health Organization (WHO) definition of anemia is <13.0 g/100 mL for men and <12.0 g/100 mL for non-pregnant women (Table 3) [95]. However, whether this definition is optimal has been a topic of discussion [95,96]. Some scientists have suggested that higher levels should be used; this is supported by a Swedish epidemiological study that used different limits/definitions in analyses of the data [97]. These authors observed an association between anemia according to the WHO definition and increased mortality (hazard ratio 2.16), but excess mortality was also observed at higher hemoglobin levels. Another study described that the severity of anemia was predictive for the underlying cause [98]: mild anemia was more frequently caused by chronic disease whereas severe anemia was more common with iron deficiency. These observations illustrate that the results from scientific studies of anemia can depend on the definition of anemia [95]. Finally, it has been suggested that the same definition with a hemoglobin level <12.0 g/100 mL should be used both for men and women (Table 3) [99]. 

It is difficult to define an optimal hemoglobin level for elderly individuals, and by strictly using the WHO definition it is not possible to take into account individual differences in hemoglobin levels [95]. An alternative strategy is to define anemia based on a decrease from previously measured hemoglobin levels, e.g., a decrease corresponding to at least 2 g/100 mL; however, for many individuals/studies it will not be possible to compare present and previous measurements.

Taken together, the observations referred to above illustrate the importance of clearly stating the definition of anemia used in clinical studies. The use of the WHO definition is important to allow comparisons between different studies, but additional analyses using/comparing different definitions may also be useful [97,98].

### 3.2. Anemia Is Common but Severe Anemia Is Uncommon in Elderly Individuals

Several previous studies have shown that anemia is common among elderly individuals in developed countries. A recent Swedish population-based study included 30,447 individuals between 44 and 73 years of age [97]. This study compared the WHO definition of anemia with alternative definitions for men/women, i.e., <14.0/<13.0 g/100 mL, <13.2/<12.2 g/100 mL, <13.0/<12.0 g/100 mL (i.e., the WHO definition) and <11.0 g/100 mL. These results are summarized in Table 3, and it can be seen that even though anemia is common in elderly individuals severe anemia (i.e., Hb < 11.0 g/dL) is uncommon. However, one should emphasize that this study included many relatively young individuals that had a relatively low mortality compared with the general population. Despite this, the results illustrate how the prevalence of anemia is highly dependent on its definition; moderate anemia is quite common whereas severe anemia is uncommon.

Other studies have demonstrated that the prevalence of anemia depends on the study population. The prevalence according to the WHO definition for elderly patients above 65 years of age is 12% for individuals living in their private homes, whereas nearly half of elderly nursing home residents and elderly patients admitted to hospital are anemic (Table 3) [99,100,101]. Some studies also describe that anemia seems to be more common for male (52%) than for female residents (32%) [100]. Finally, the prevalence of anemia also depends on age [99,101]: a previous study described that 11.0% of men and 10.2% of women 65 years or older were anemic, but that the prevalence of anemia rose rapidly to more than 20% at 85 years of age or older [95,99].

### 3.3. Causes of Anemia in Elderly Individuals

The cause of anemia in elderly individuals was investigated in a prospective American study that included 190 patients above 65 years of age [102]. All the patients were referred to hematological out-patient wards and diagnosed with anemia according to the WHO criteria. They all lived at home without help, and the exclusion criteria were known hematological disease, expected survival <3 months and renal failure requiring dialysis. These individuals were compared with a matched control group without anemia. All participants were interviewed and a clinical examination was performed as was a blood sample examination, including peripheral blood cell counts, examination for iron deficiency and levels of folic acid, cobalamin, thyroid-stimulating hormone, erythropoietin and creatinine, with an estimation of the glomerular filtration rate. For most individuals protein electrophoresis (performed for 86% of the patients) and microscopy of peripheral blood smears were performed, whereas bone marrow examination was performed only for a minority. If an individual had more than one cause of anemia they were classified according to the main cause. The following observations were made:Six percent of the patients were diagnosed with a hematological malignancy, the most common being myelodysplastic syndrome (MDS), which was the suspected cause for 16% of the patients.Eleven percent had a non-hematological malignancy.Twelve percent had iron deficiency, but only a minority of these patients had microcytic anemia and for many patients the hemoglobin level did not normalize in response to iron supplementation. Iron deficiency was thus a possible contributing cause of anemia for many of these patientsRenal failure was the cause of anemia for 4% of the patients.Long-lasting inflammation was the cause for 6% of the patients.Anemic patients and controls did not differ with regard to pharmacotherapy.For 35% of the individuals the cause of their anemia was not found. None of these individuals had hemoglobin levels below 9 g/dL (i.e., they probably did not require regular erythrocyte transfusions) and there was no association with ethnicity, age or sex. However, many of these patients had increased erythrocyte sedimentation rates and ferritin levels, i.e., they had systemic signs of an acute-phase reaction.

Many elderly individuals with anemia are probably handled by general practitioners without a diagnostic follow-up at a hematological out-patient ward, and the present patient population therefore represents a selected group compared with the general population of elderly individuals. A relatively large group of these elderly patients are characterized by an unknown cause of anemia, a moderate decrease in the hemoglobin level and systemic signs of inflammation. The cause of anemia in elderly individuals has also been investigated in other studies [99,102,103], and the overall results show that a relatively large number of elderly patients with anemia has an unknown cause after a limited evaluation based on clinical examination and blood samples (Table 4).

What are the possible causes of anemia for the large group of patients with anemia of an unknown cause? First, one possibility is low-risk MDS with moderate anemia as the only sign of the disease; these variants of MDS can be difficult to diagnose even after repeated examinations. Many of these patients have macrocytic anemia, and the relatively short survival of anemia patients with increased mean corpuscular volume (MCV) in a large population study is consistent with this hypothesis; in this study, macrocytic anemia was rare and associated with a higher mortality than normocytic and microcytic anemia [97]. Second, pharmacotherapy may also be a possible cause, e.g., renin–angiotensin inhibitors are commonly used in patients with cardiovascular disease and can be associated with anemia [104]. Third, inflammation/inflammaging may be the cause of anemia in these patients [102,105]. In a previous study only including individuals above 65 years of age, it was observed that (independent of age, sex and hemoglobin) the number of elevated proinflammatory markers (CRP, IL6, IL1β and TNFα) was associated with progressively higher erythropoietin levels in nonanemic individuals but with decreased erythropoietin in anemic participants [105]. These last observations were consistent across different causes of anemia, and the hemoglobin threshold at which the association between inflammation and erythropoietin reversed was approximately hemoglobin 13.0 g/100 mL. These observations suggest that inflammaging (i.e., all individuals were above 65 years of age) is associated with a pre-anemic stage of high erythropoietin followed by a decrease in erythropoietin and the development of anemia. To conclude, in our opinion the large group of elderly with anemia of an unknown cause is most likely a heterogeneous group where the anemia can be caused by preleukemic MDS, pharmacotherapy, inflammaging and probably other causes.

### 3.4. The Diagnostic Evaluation of Anemia in Elderly Patients

The large group of individuals with an unknown cause of anemia reflects that the diagnostic evaluation was limited in these previous epidemiological studies. A recent review has suggested that the initial laboratory evaluation of anemic elderly patients should include the samples listed in Table 5 [106]. This list is more extensive than the evaluation used in previous epidemiological/clinical studies, and one would therefore expect the group of patients with unexplained anemia to decrease if this diagnostic strategy is used. In our opinion this is a reasonable diagnostic compromise.

The difficult question is how extensive the additional diagnostic evaluation should be if the cause of the anemia is still unknown after this initial examination. First, this group may include patients with androgen deficiency [107,108], vitamin D deficiency [109] or altered erythropoietin homeostasis [105,108,110]. Second, an additional evaluation may become necessary to establish the diagnosis of early vitamin B12 or folic acid deficiency. Third, the initial laboratory evaluation may suggest gastro-/colonoscopy or an ultrasound examination of the abdomen/kidneys. Finally, in our opinion the most difficult question is whether a more extensive bone marrow examination is justified, i.e., bone marrow aspiration, bone marrow biopsy, cytogenetic analysis and/or molecular genetic analyses. As will be discussed later, anemia was associated with increased mortality in the prospective NHANES III study, and 17% of the anemic patients in this study had features suggesting MDS or another myeloproliferative disease (e.g., unexplained MCV increase, additional cytopenia) [99]. An Israeli study described that 15% of cognitively impaired hospitalized patients with unexplained cytopenia had evidence of MDS [111], an American study described that mutations could be detected for 40% of patients with idiopathic cytopenia of uncertain significance when using a 22-gene mutation panel [112] and a British study described a high percentage of MDS-associated mutations in patients with nondiagnostic marrow biopsies [113]. Molecular genetic analyses are now available and in a recent review the authors concluded that clinical testing for mutations in hematopoietic cells is reasonable in cases of unexplained anemia of older patients, especially if additional cytopenias are present [114]. Other authors have suggested that bone marrow evaluation should only be considered for patients with an expected survival of at least three months [106]. In our opinion the best justified recommendation is that a bone marrow evaluation (including mutational analyses) should be considered for individual patients after a careful evaluation that includes the burden of the procedure, possible therapeutic consequences, life expectancy and the burden of the anemia.

### 3.5. Anemia as a Prognostic Parameter in Community-Living Elderly Individuals

As described in detail in Table 6, anemia is common for elderly individuals (approximately 10% of persons above 65 years of age) and the incidence rate increases with increasing age, but severe anemia with Hb below 11 g/100mL is seen only for 2% or less of individuals depending on the study population (Table 2, Table 5 and Table 6) [97,98,99,115,116,117,118,119,120,121,122,123,124,125,126,127]. The incidence of anemia seems to depend on race and is higher in black Americans [99,116]. The hemoglobin level associated with increased mortality also seems to depend on race: for white non-Hispanic Americans hemoglobin levels below the WHO cut-off is associated with increased mortality, whereas the mortality is increased for black American and Mexican Americans with levels lower than 1 g/100 mL below the WHO cut-off [116]. Below these cut-off points a five-year survival of 40–45% was observed, whereas individuals without anemia had a survival exceeding 80% [116]. Finally, the association between anemia and increased mortality as well as hospitalization is also observed when only including patients without prevalent disease in the studies [117].

Several prospective studies have demonstrated significant clinical effects of anemia:Anemia has a negative impact on survival, but this impact seems to differ between subtypes based on the relative risk in the order nutritional > chronic kidney disease > inflammation > unknown cause [118,119,120].Mild anemia is also associated with reduced physical performance, muscle strength, cognition and quality of life [117,127].Anemia is associated with an increased risk of depressive symptoms [128].Anemia with chronic inflammation is associated with autoimmune disease but also with cancer [115].

Thus, elderly anemic patients often have complex clinical problems that have to be considered when evaluation and possible treatment of the anemia is considered.

### 3.6. Anemia as a Prognostic Parameter in Nursing Home Residents

Anemia is more common among elderly nursing home residents than in community-living elderly individuals; this has been demonstrated in several studies from different countries, including those representative studies summarized in Table 7 [100,121,122,123,124,125,126]. Several studies have shown that more than 50% of residents have anemia according to the WHO definition [95]. The most important causes of anemia in these patients are nutritional factors, renal failure and chronic inflammation (Table 4), and anemia becomes more frequent with increasing age [98,99,126]. This is similar to community-living elderly (see Table 6 and Table 7). A large subset of the anemic patients has an unknown cause after the initial routine evaluation based on clinical examination and blood sample analyses (Table 4), and hematological malignancies were not found to be a frequent cause of anemia in elderly patients in any of the studies described in Table 7. Furthermore, it should be emphasized that anemia is frequently multifactorial [129].

Some studies suggest that the frequency of anemia also differs between men and women for nursing home residents [100,123]. Furthermore, anemia is also dependent on race, and several studies have described higher frequencies in black residents [122,124]. Finally, several studies have described an association between anemia and survival, and the more severe the anemia the stronger the prognostic impact [100,125]. Some studies also suggest that this association is strongest for men [100].

Anemia seems to be a part of a more complex clinical situation with reduced function for these patients. First, even mild anemia and low normal levels are often associated with lower muscle strength, physical function and mobility [129]. Patients with anemia below 11 g/100 mL also have significantly decreased scores for activities of daily life and quality of life [130]. Second, even though at least one study has shown no associations between anemia and decreases for community-living elderly individuals [131], such an association has been observed for nursing home residents [132,133]. Third, frailty has been defined as a medical syndrome characterized by decreased physiological reserve and increased vulnerability, and frailty seems to be a predictor for nursing home placement of elderly community-dwelling individuals [134]. Anemia may therefore be only a part of a complex physiological reduction. Finally, being underweight is also a risk factor (in addition to anemia?) for mortality in elderly nursing home residents [135], and anemia is important for the quality of life of cancer patients [136]. Taken together, these studies show that anemia will often be a part of a complex clinical situation for elderly individuals, including other factors that are also associated with mortality.

## 4. Causes of Mortality in Elderly Individuals with Anemia

Anemia in the elderly is very heterogeneous with regard to its etiology, and it is therefore not surprising that different causes contribute to the increased mortality.

### 4.1. Increased Mortality from Stroke

The impact of anemia on the mortality of patients with stroke was addressed in a recent meta-analysis based on 13 cohort studies including 19,239 patients [137]. Anemia was associated with an increased risk of mortality in stroke. This prognostic impact of anemia is possibly seen for patients with less severe stroke in particular [138]. However, the large meta-analysis was based on studies that also included younger patients, whereas the registry study by Barlas et al. [139] included 8013 patients with a mean age of 77.8 years. In this last study anemia was present at admission in 24.5% of the patients, and increased mortality was observed both for men and women with ischemic stroke. A more recent study by Barlas et al. [140] suggested that microcytic and normocytic anemia differed with regard to mortality and disability after stroke.

### 4.2. Increased Mortality from Heart Disease: Studies in Patients with Chronic Heart Failure

Several studies have investigated the association between anemia and mortality in patients with chronic heart failure:One study included 6159 outpatients with stable chronic heart failure [141]. The prevalence of anemia was 17.2% (median age: 69 years for anemic versus 65 years for nonanemic); after six months 43% of these anemic patients at baseline had normalized Hb levels, whereas 16% of the nonanemic patients had developed anemia. After a mean follow-up of 3.9 years the mortality was higher both for patients with persistent anemia (58% vs. 31%, *p* < 0.0001) and incident anemia (45% vs. 31%, *p* < 0.0001) compared with nonanemic individuals at six months.A meta-analysis based on 153,180 heart failure patients included 37.2% anemic patients [142]; after a follow-up of at least six months the mortality was 46.8% for anemic and 29.5% for nonanemic patients. Lower baseline Hb was associated with higher mortality. These observations were also supported by another meta-analysis [143]: the patients in 10 of the 20 included reports had a mean age above 60 years, and an association between anemia and more severe heart failure was observed.The study by Kosiborod et al. [144] included 2281 patients aged 65 years or older with heart failure. This study showed that elderly patients with heart failure and anemia had higher one-year mortality.

To conclude, anemia is associated with increased mortality for patients with heart failure, including elderly patients.

### 4.3. Nutritional Defects and the Role of Iron Deficiency in Patients with Heart Failure

The possible role of iron deficiency alone in heart failure has been investigated in several studies [145]. By using a multivariable hazard model, iron deficiency, but not anemia, was found to be a strong and independent predictor of mortality in a study of 1506 patients with chronic heart failure [146]. These authors defined iron deficiency as ferritin <100 μg/L or ferritin 100–299 μg/L together with a transferrin saturation of <20%, and this was present for 753 patients. Thus, iron deficiency was common, was associated with the severity of heart failure and was an independent prognostic marker. The possible importance of iron deficiency is also supported by two recent clinical studies describing a reduced rehospitalization rate after iron supplementation for patients with heart failure [147,148], but none of these studies could detect any effect of iron supplementation on survival. A third study could not detect any effect of iron supplementation on physical capacity either [149]. Thus, the overall results suggest that iron supplementation has only a limited effect on patients with heart failure and iron deficiency, whereas the association between anemia and mortality has been detected in several large studies.

A previous study could not detect any association between vitamin B12 or folate deficiency and mortality for patients with chronic heart failure [150].

### 4.4. Anemia in Patients with Cancer

Anemia is a common symptom of cancer, and 20–60% of patients with cancer have anemia at their initial diagnosis [151]. This is often referred to as the anemia of cancer, but it should be emphasized that anemia in cancer patients can be multifactorial and that possible additional contributing factors can be nutrition, inflammation, bleeding with iron deficiency or extensive bone marrow infiltration of malignant cells [151,152]. Although very few epidemiological studies of anemia in elderly patients have investigated how undiagnosed malignant disease contributes to the increased mortality of these patients, this is suggested by several observations. First, a Korean study of 10,114 elderly and apparently healthy individuals (mean age of 64 years) described an increased risk in all-cause mortality and cancer-related mortality (especially lung cancer) in men but not in women [153]. Second, a recent population-based cohort study including 138,670 individuals aged 18–93 years investigated the impact of anemia on survival [154]. An association between anemia and survival was observed especially for elderly patients (i.e., above 80 years of age). This adverse effect on survival was associated with both anemia and signs of chronic inflammation, whereas the survival was higher for patients with nutrient deficiencies and anemia of an unknown cause. As will be discussed below, the anemia of cancer is associated with inflammation. Third, unexplained anemia can be the first sign of low-risk MDS, but small studies including relevant diagnostic procedures have concluded that MDS could be diagnosed only for a small minority (i.e., less than 15%) of patients with unexplained anemia after a limited non-invasive evaluation [155]. Finally, iron deficiency can also be associated with bleeding from an undiagnosed gastrointestinal tumor, and this is one of the reasons why upper and lower gastrointestinal endoscopy have been recommended for patients with unexplained iron deficiency anemia [156]. This clinical strategy is also supported by clinical experience showing that even elderly patients with recurrent iron deficiency anemia may have a cause of iron deficiency anemia that can be treated [157,158]. Taken together, these observations strongly suggest that undiagnosed malignancy can be a cause of anemia in elderly individuals, and that these cancer patients can hide among patients with iron deficiency, anemia with chronic inflammation and anemia with an unknown cause after a limited noninvasive diagnostic evaluation.

Anemia is a common symptom of cancer and is often referred to as the anemia of cancer [151]. The cytokine-induced inhibition of erythropoiesis is regarded as an important mechanism for the development of anemia in cancer patients, TNFα in particular but also other inflammatory regulators, including IL1β, IL6, IL10 and IFNγ, probably contribute [151,159]. As discussed in previous reviews, these mediators have direct inhibitory effects on erythroid progenitors; the mechanisms differ between the various cytokines and include the inhibition of proliferation and differentiation, the downregulation of erythropoietin receptors and the induction of apoptosis through the Fas pathway [151]. However, proinflammatory cytokines may also have indirect effects on erythropoiesis, including the inhibition of renal erythropoietin production and the production of hepcidin leading to iron retention in macrophages and decreased dietary iron absorption [151,159]. Finally, uncommon causes that contribute to anemia in cancer can be (i) immune-mediated hemolytic anemia as a paraneoplastic disease, (ii) microangiopathic anemia caused by extensive cancer metastases with pathological microvessels in the tumor, (iii) microangiopathy caused by cancer-associated thrombotic thrombocytopenic purpura, (iv) hemolytic uremic syndrome or (v) anemia as a part of cancer-associated coagulopathy with disseminated intravascular coagulation [151].

Anemia and/or blood transfusions can be adverse prognostic parameters in patients with cancer [151,160]. Preoperative anemia can be an independent risk factor associated with both survival and relapse risk in cancer patients [161,162].

The treatment of cancer-associated anemia is outside the scope of this article, but it has been addressed in several recent articles [152,163].

### 4.5. Increased Mortality of Anemic Patients after Surgery

A large meta-analysis investigated the association between preoperative anemia and mortality after surgery [164]. This analysis included not only studies of elderly patients: 24 eligible studies were identified and these studies included 959,445 patients, of which 371,594 patients were anemic. Anemia was then associated with increased mortality (odds ratio of 2.90/*p*-value < 0.001), acute kidney injury (3.75/<0.001) and infection (1.92/00.01). These findings were similar for the cardiac and non-cardiac surgery patients, but anemia was also associated with stroke for the cardiac surgery patients. Thirteen of these studies used the WHO definition of anemia, and the association between anemia and postsurgery mortality remained when the analysis was restricted to these patients. It seems justified to conclude that presurgery anemia reflects a more complex clinical situation associated with increased postsurgery mortality.

We have identified four studies investigating the postoperative mortality for elderly patients. First, one study based on a national prospective database included 31,857 elderly patients above 65 years of age undergoing an elective vascular operation [165]. Forty-seven percent of these patients had anemia, and the anemic patients had increased 30 days postoperative mortality (2.4% versus 1.2%, *p* < 0.0001) and cardiac event rate (2.3% versus 1.2%, *p* < 0.0001) compared with the nonanemic patients. The mortality was highest for the patients with severe anemia. Second, another study included 310,311 patients aged 65 years or older who underwent major noncardiac surgery [166]. This study also observed an increased mortality associated with anemia. Finally, a large study investigated the association between anemia at presentation and postoperative mortality for elderly patients with a mean age of 79.2 years [167]. At presentation, 65% of these patients were anemic, and anemia was then associated with an increased odds ratio for mortality (1.3/*p* = 0.004); there was no significant association between anemia and myocardial infarction or cerebrovascular events. Anemia seems to be an indicator of poor general health and thereby increased mortality after surgery for many elderly patients [168].

### 4.6. Summarizing Comments: Anemia, Inflammation and Mortality

Anemia is common in elderly individuals, and the increased mortality in elderly patients with anemia is multifactorial. Although anemia and inflammation are present together in many patients, several studies suggest that anemia has an effect on mortality that is independent of the concomitant inflammation. An adverse prognosis due to common disorders and comorbidities (including cardiovascular disease and cancer) seems to be most important and definitely more important than an increased frequency of uncommon diseases (e.g., hematological malignancies).

## 5. Inflammation in Aging

### 5.1. CRP as a Marker of Inflammation: Its Structure and Function

CRP is an acute-phase protein and is used as a marker of inflammation both in epidemiological studies and in routine clinical practice [169]. As described in several previous reviews, CRP exists in several isoforms [170,171,172,173,174,175,176]. It is synthesized as monomers; the pentamer is thereafter formed in the endoplasmic reticulum where it is also stored and from where it is released slowly during the non-inflammatory baseline situation. Thus, native CRP is a pentameric protein, but it can also be detected extracellularly as a monomer (206 amino acids and a molecular weight of 23 kDa) formed by the irreversible dissociation of the released pentameters. The pentamer is rapidly released in response to increased levels of proinflammatory cytokines. Finally, CRP can also form fibril-like structures, decamers and possibly trimers as well as tetramers; various CRP peptides can also mediate biological effects [177].

The acute-phase reaction is characterized by an acute increase in the systemic (i.e., serum/plasma) levels of several proteins in response to inflammation, infection or tissue injury [86]. However, it can also be seen in chronic diseases or be a chronic or long-lasting response [169]. The reaction is regarded as a response that is induced by cytokines produced at inflammatory sites; IL6 is then an important stimulator together with other members of the IL6 family, IL1β, TNFα, IFNγ, TGFβ and IL8/CXCL8 [86]. Several of these cytokines/chemokines are involved in the regulation of CRP gene expression, including TNFα, IL6, IL8/CXCL8 and CCL2 [169]. Thus, even though CRP is only one out of several acute-phase proteins, it should be regarded as an immunoregulator that reflects and integrates intercellular signaling mediated by several proinflammatory mediators.

The release and immunoregulatory functions of CRP have also been described in previous reviews [86,169]. Briefly, the native CRP isoform is mainly released by hepatocytes but can also be released by smooth muscle cells, macrophages, endothelial cells, lymphocytes and adipocytes. Its biological effects include [86,169,178,179,180,181]:Monocytes/macrophages: These cells can be polarized by CRP towards the proinflammatory M1 phenotype with increased phagocytosis and cytokine release, inhibited chemotaxis and altered metabolism with increased LDL uptake.Dendritic cells: CRP seems to be an important regulator of dendritic cell functions and can activate monocyte-derived dendritic cells [178,179,180].T cells: Indirect stimulation/modulation of T cell activation through the effects on dendritic cells [179,180].Neutrophils: The functional CRP effects depend on the biological context and can be decreased activation, inhibition ofchemotaxis and/or stimulated phagocytosis [181].Endothelial cell activation.Thrombocytes: Inhibition of activation, trafficking and aggregation.Complement activation.

To conclude, CRP is a common target that integrates information from several upstream events/immunoregulators, but at the same time CRP itself is an important immunoregulator that influences the function of several immunocompetent cells. CRP should therefore be regarded as a key point in the network of soluble immunoregulators.

### 5.2. Inflammaging and Systemic CRP Levels in Elderly Patients

Aging is associated with the accumulation in many tissues of senescent cells with a secretory phenotype; these cells can release proinflammatory cytokines, chemokines and other mediators that modulate their microenvironments [46,182]. Animal models suggest that this increased release is associated with the increased activation of JAK-STAT pathways [183]. This chronic state of low-grade inflammation also seems to be reflected in the increased systemic levels of several proinflammatory cytokines/markers, including IL1β, IL1Rα, IL6, TNFα and IFNγ [184,185]. These mediators may then influence various physiological systems and contribute to the complex process of aging and the clinical situation of many elderly people, including altered hematopoiesis and neurological functions [186]. This process is often referred to as inflammaging [187]. These mediators are also initiators and drivers of the acute-phase reaction [169], and these effects may therefore at least partly explain why systemic CRP levels should be regarded to reflect the process of inflammaging. Inflammaging is probably caused by age-dependent functional alterations in various immunocompetent cell types, including both the innate and adaptive immune system [186,188].

The process of inflammaging is not necessarily associated with disease development. We recently investigated the CRP levels in a group of 85 healthy allogeneic stem cell donors [189]. After a careful evaluation none of these donors showed any signs of disease, and they were all regarded as acceptable stem cell donors. However, a subset of these stem cell donors showed increased CRP levels, and these increased levels were observed especially for elderly donors. The CRP levels of this donor subgroup were further increased by stem cell mobilization by G-CSF, and our studies suggest that IL6 and possibly other members of the IL6 family can influence this systemic low-grade inflammation associated with aging.

### 5.3. Inflammation-Induced Modulation of the Hematopoietic Stem Cell Niche

The various hematopoiesis-supporting stromal cells of the stem cell niches express a wide range of cytokine/chemokine receptors as well as pattern recognition receptors [190]. These receptors recognize ligands that are generated locally or systemically; their cellular functions can thereby be modulated and a new or second wave of mediators released locally in response to a systemic response/reaction. This second wave also includes prostaglandins and enzymes, e.g., prostaglandin E2 and nitric oxide synthase [191]. These locally released mediators may thereafter modulate the trafficking of hematopoietic/immunocompetent/immunomodulatory cells to the bone marrow [192,193]. Another example is the endothelial cells: these cells express multiple pattern-recognizing receptors, and the stimulation of Toll-like receptor 4 induces G-CSF release whereas proinflammatory cytokines (including IL6 released by MSCs) stimulate the release of GM-CSF by endothelial cells and modulate the endothelial cell responsiveness to proinflammatory cytokines [190,194,195,196,197]. The endothelial cells thus contribute to the translation of proinflammatory signals into the regulation of hematopoiesis. Finally, cell–cell contact through adhesion molecules or membrane-expressed ligands that induce intracellular signaling are also involved in this translation [190,196]. A complete review of the molecular mechanisms between these stromal cells as well as between stromal and hematopoietic cells is beyond the scope of this review, but our examples clearly show that inflammation affects the functional phenotype of various stromal cells in the stem cell niches and thereby indirectly modulate hematopoiesis.

### 5.4. Inflammation, Aging, Disease and Mortality

Several studies suggest that inflammation (i.e., increased CRP and/or cytokine levels) is associated with an increased risk of future cardiovascular events [198,199,200,201,202,203]; as discussed in detail in a previous review, this is also true for elderly patients above 65 years of age (Table 8) [204]. First, Tracy et al. [198] described an association between high CRP levels and future coronary heart disease: this association was seen especially for women with subclinical cardiovascular disease who had a cardiovascular event within one year. The association was also significant when analyzing myocardial infarction alone. Second, the two studies by Cesari et al. [199,200] also described significant associations between cardiovascular disease and CRP levels, and in addition they described associations between IL6 and TNFα levels with cardiovascular disease that seemed to be stronger than the association with CRP. However, another study concluded that CRP was less useful as a prognostic marker in elderly individuals; the associations between CRP and cardiovascular health did not reach significance after adjustment for several other cardiovascular risk factors, and there was no statistical evidence for a gender interaction either [204]. Other authors have described an association between inflammation and cardiovascular events also when using a composite indicator of inflammation [205]. Third, Makita et al. [201] detected a gender difference with a significant association between carotid plaque score and high CRP levels only for men. Finally, even though CRP may not be an independent risk factor, it seems to be a part of a clinical high-risk inflammatory phenotype with complex interactions between several risk factors in elderly individuals, including smoking, diabetes, hypertension, body mass index, lipid metabolism and inherited differences in the regulation of inflammation [202,205,206,207,208,209,210]. Gender differences with regard to the impact of CRP have been described only in some studies, but differences between men and women with regard to the associations between CRP levels and cardiovascular risk would not be unexpected because associations between CRP levels and male sex hormone levels have been described [211]. Taken together, these observations suggest that there is an association between inflammaging/CRP increase and the risk of clinical cardiovascular disease. However, as concluded by a more recent study, there may not be a causal association: increased CRP levels seem rather to reflect hidden inflammatory activity that is strongly associated with all-cause and not only cardiovascular mortality [212].

A recent systematic review of 23 cohort studies analyzed the associations between blood biomarkers and mortality [213]. These authors included studies with a mean age between 50 and 75 years at baseline. The meta-analysis of mortality risk showed significant associations not only with cardiovascular mortality but also with all-cause mortality and cancer mortality. Twenty biomarkers showed associations with mortality risk, and among them were several markers of inflammation/acute-phase reaction, including total white blood cell count, circulating neutrophil granulocytes, erythrocyte sedimentation rate, fibrinogen and TNF receptor II.

## 6. Summarizing Discussion

In this review we have described the biological background and clinical aspects of anemia and inflammation in elderly individuals. Both anemia and inflammation are common in the elderly and in our opinion they should be regarded as related, although only partly overlapping processes, with regard to pathogenesis and prognostic impact.

### 6.1. Anemia and Aging

Aging is characterized by several cellular hallmarks, including genetic instability, telomere shortening, loss of proteostasis, deregulated nutritional sensing, mitochondrial dysfunction, cellular senescence, stem cell exhaustion and altered cellular communication [2]. These factors will also influence hematopoiesis and hematopoietic stem cells in elderly individuals. The biological context of anemia as well as inflammation will therefore be different from that of younger individuals. The ageing in hematopoiesis seems to be caused by several mechanisms and complex interactions between aging-associated alterations in hematopoietic cells (including hematopoietic stem cells) and alterations in hematopoiesis-supporting bone marrow stromal cells [14,60,78]. The inflammation itself can also be a part of the aging process, and is then referred to as inflammaging [185,187]. It should also be emphasized that inflammaging occurs in the context of complex age-associated alterations in the innate and adaptive immune systems [186]. Many of the disorders that can cause anemia and/or signs of inflammation in elderly individuals are quite common (Table 3, Section 3.3), but little is known about whether or how the treatment of such diseases should be modified in elderly individuals due to differences in the biology or pathogenesis of these disorders caused by the effects of the aging of the involved cells, e.g., aging-associated effects in immunocompetent cells for patients with autoimmune diseases.

### 6.2. Carcinogenesis and Cancer Treatment in Elderly Individuals

Age-associated differences also seem to be reflected in the development and biological characteristics of malignant diseases in elderly individuals, e.g., hematological malignancies [94]. This has been demonstrated by proteomic studies on favorable prognosis AML (i.e., favorable genetic abnormalities) where the proteomic profiles differ between elderly and younger patients with the same cytogenetic abnormalities [96]. In our opinion the same is probably true for other malignancies: precarcinogenic aging-associated differences probably remain after carcinogenesis in malignant cells.

Another aspect is the toxicity of anticancer therapies in the elderly. Age-dependent differences are probably the explanation as to why several forms of anticancer treatment cannot be used in elderly individuals. Toxic effects and especially hematological toxicity, often dose-limiting for anticancer treatment [214], as well as immune-related toxicity are more severe in allogeneic stem cell transplantation both when using elderly stem cell donors [215,216,217] and for elderly recipients [218,219]. The same may be true for radiation therapy, and in our opinion an increased mortality similar to the mortality in orthopedic surgery [164,165,166] would also be expected for cancer surgery.

Signs of inflammation and an acute-phase reaction is associated with an adverse prognosis and decreased survival in many cancer forms [169], but at the same time the use of various forms of immunostimulatory therapies (e.g., checkpoint inhibitors) has improved the prognosis/survival for many cancer patients [220]. The explanation for this apparent discrepancy is possibly that the acute-phase reaction can be associated with local macrophage infiltration that enhances tumor growth through the release of growth factors (e.g., the stimulation of local angiogenesis) despite the concomitant induction of a systemic acute-phase reaction [221], whereas effective anticancer immunotherapy targets and enhances anticancer T cell responses.

### 6.3. Inflammation and Anemia

CRP is often used as a sign of inflammation and inflammaging, and cohort studies have described associations between CRP levels and overall cardiovascular and cancer mortality. However, it should be emphasized that several mechanisms can induce the acute-phase reaction, and increased CRP levels can be associated with different soluble mediator serum/plasma profiles. This last aspect is illustrated by two recent studies on cancer patients: inflammatory markers are associated with a prognosis in both head and neck squamous cell carcinoma as well as renal cancer, but the prognostic impact of systemic levels of individual proinflammatory mediators seem to differ between the two groups [222,223]. In our opinion it will therefore be important to investigate the systemic inflammation-associated mediator profile and not only the CRP levels to further characterize individual differences as well as differences in the prognostic impact of inflammaging. The observation that systemic levels of single proinflammatory cytokines show stronger associations than CRP levels to cardiovascular health further support this hypothesis [200].

For several reasons anemia in the elderly and inflammaging should be regarded as related (i.e., events with the same upstream initiator) and/or complicating events with only overlapping but not identical molecular pathogenic mechanisms. First, patients with signs of inflammation are regarded as a specific subset of elderly anemia patients (Section 3.3). Second, proinflammatory cytokine responses involving IL1, TNFα and IL6/IL6 family members can induce the systemic acute-phase response, and the same cytokines can also contribute to the aging effect on normal hematopoiesis as well as to the development of anemia due to a specific cause, e.g., they are important in the development of cancer-associated anemia (Section 2.4 and Section 6.3). Third, a subset of elderly patients with anemia is characterized by inflammation and increased CRP levels, but it should also be emphasized that anemia seems to have an impact on survival that is independent of the CRP level [120]. Finally, both anemia and inflammation are associated with the same caused of mortality, e.g., cardiovascular death and possibly cancer-related mortality (Section 4 and Section 5.4).

In our opinion it seems likely that there is partly an overlap in the biology/pathogenesis and clinical impact of anemia and inflammaging. However, signs of inflammation in patients with anemia are not necessarily caused by inflammaging but can alternatively be caused by a specific disease that initiates two (partly) independent events, e.g., in malignancies or autoimmune diseases [115]. The association between anemia and inflammation seems to be relatively weak, because despite the significant correlations between CRP and hemoglobin levels in anemic patients [100] the prognostic impact of anemia seems to be significant even after correcting for CRP [120]. A better understanding of the biological/molecular mechanisms behind the prognostic impact of anemia/inflammation/inflammaging will also be a necessary scientific basis when considering possible therapeutic interventions in patients with anemia and/or inflammation.

### 6.4. Anemia of an Unknown Cause

Several population-based studies of elderly individuals with anemia describe a relatively large group with anemia of an unknown cause (Table 3). Some previous authors have emphasized that these patients have been inadequately investigated when taking into account the diagnostic tools that are now available (see Section 3.3). In our opinion the word inadequate is misleading: one should rather refer to these patients as having an unknown cause after a limited diagnostic evaluation. Furthermore, a recent study suggests that the cause of anemia is unknown for many patients even after a more extensive laboratory work-up [224]. The extent of the diagnostic evaluation should, in our opinion, be individualized and based on an overall clinical evaluation. For many elderly patients an extensive diagnostic evaluation will not have therapeutic or prognostic consequences, and a limited diagnostic evaluation may therefore be relevant.

## 7. Conclusions

Aging is associated with the intrinsic (i.e., age-associated alterations of hematopoietic cells, including the stem cells) and extrinsic modulation of hematopoiesis caused by age-associated alterations of hematopoiesis-supporting bone marrow stromal cells. Furthermore, aging-associated modulation has also been described for most subsets of immunocompetent cells. These aging-associated alterations are reflected by the frequent detection of both anemia and signs of inflammation in elderly individuals. Anemia and inflammaging should be regarded as related, as they at least partly reflect the same biological mechanisms (e.g., increased levels of several proinflammatory mediators). Both anemia and inflammation are associated with increased mortality: the background for this decreased survival is probably multifactorial but seems to include increased cardiovascular mortality. Additional biological characterization of the molecular mechanisms behind anemia and inflammation is necessary to improve the clinical handling of individual patients, and the handling of these elderly patients should, in our opinion, be individualized and based on the overall clinical situation.

## Figures and Tables

**Figure 1 jcm-11-00706-f001:**
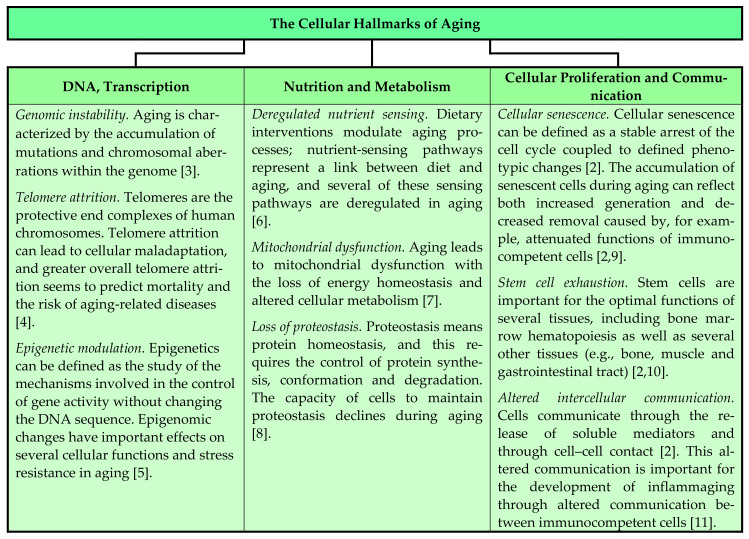
The biological hallmarks of aging: an overview and summary of biological characteristics [2,3,4,5,6,7,8,9,10,11].

**Figure 2 jcm-11-00706-f002:**
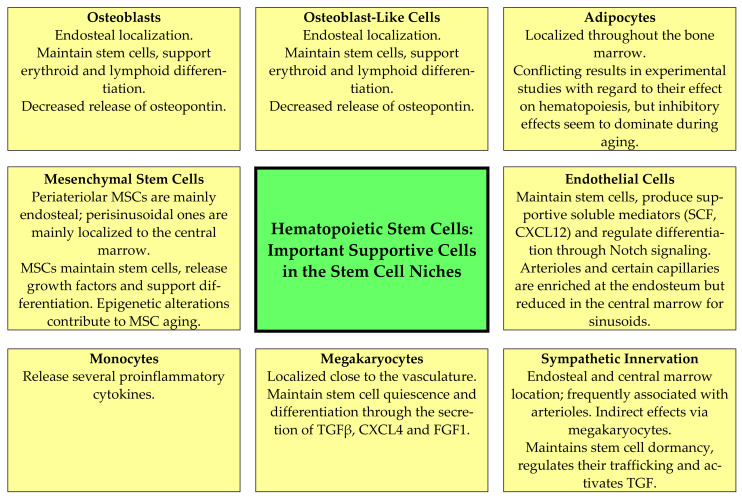
The bone marrow stem cell niches. The figure gives an overview of important hematopoiesis-regulating members of the various stem cell niches and their main regulatory effects/mechanisms on normal hematopoiesis A detailed discussion of each cell type with corresponding references are given in Section 2.3 (abbreviations: CXCL, C-X-C motif ligand; FGF1, fibroblast growth factor; SCF, stem cell factor; and TGF, transforming growth factor).

**Table 1 jcm-11-00706-t001:** A summary of important cell-intrinsic mechanisms involved in aging of hematopoietic cells; for a detailed review and discussion with references we refer to Section 2.2.

*Genetic:* Genetic instability with accumulating DNA damage and clonal hematopoiesis; this is due to the altered function of several mechanisms involved in genomic maintenance/DNA repair.
*Epigenetic:* Epigenetic modulation with altered chromatin organization, posttranscriptional histone modulation and DNA methylation; transcriptional regulation is thereby altered.
*Polarity:* Reduced cytoplasmic and nuclear polarity, reduced ability of asymmetric cell division.
*Metabolism:* A shift to higher oxidative metabolism, altered proteostasis due to reduced autophagy and reduced activity of the proteasome system, reduced endoplasmic reticulum stress response with the accumulation of misfolded or damaged proteins.
*Senescence, signaling and communication:* Accumulation of cell-cycle-arrested senescent cells, altered intercellular communication and intracellular signaling possibly involving auto- and paracrine circuits, reduced regenerative capacity of hematopoietic stem cells.

**Table 2 jcm-11-00706-t002:** Extrinsic mechanisms for the aging of normal hematopoiesis: a summarizing overview of important mechanisms behind the contribution of various stromal cells to the aging of normal hematopoiesis (for additional information, see the more detailed review/discussion with corresponding references for each cell type in Section 2.3).

Stromal Component	Important Effect of Aging
MSCs [19,52,53,54,55,56,57,58,59,60,61]	Maintained or increased central MSCs with decreased hematopoietic growth factor production; loss of periarteriolar MSCs. Decreased bone formation and increased adipogenesis; MSC aging is also characterized by increased senescence (including altered mediator secretion) and epigenetic modifications.
Osteoblastic cells [19,55,62,63]	Decreased number of osteoblasts, increased differentiation in the direction of adipocytes. Decreased numbers of osteoblasts represent decreased support of lymphopoiesis. Decreased osteopontin release; this cytokine can attenuate the aging-associated phenotype of hematopoietic stem cells.
Osteoblasts [27,64]	Decreased number and release of osteopontin, decreased osteoblast number and thereby reduced support of lymphopoiesis.
Adipocytes [19,65,66,68,69]	Increased number of adipocytes during aging. The effects of adipocytes on normal hematopoiesis depend on the biological context. Hematopoiesis is often suppressive, and the release of adiponectin probably contributes to this inhibition. However, adipocytes or a subset of them also seem to facilitate regeneration after chemotherapy or irradiation through their release of SCF.
Endothelial cells [19,53,54,55,56,57,72,73]	Loss of certain capillaries and arterioles, decreased release of SCF and CXCL12 in addition to decreased expression of the Notch ligand Jagged1. Increased or unaltered endothelial cell pool, altered microvascular function with increased vascular leak.
Perivascular cells [17,70]	Aging of these cells is associated with reduced number of these cells and thereby reduced release of soluble stem cell-supporting mediators, e.g., SCF.
Sympathetic innervation [19,54,57,74]	Loss of sympathetic innervation in aging; this leads to expansion of medullary MSCs with decreased supportive effect, reduces the number of arterioles and increases the hematopoietic stem cell number.
Megakaryocytes [18,19,54,57,79,80,81,82,83]	Increased number and TGFβ release in aging.

Abbreviations: CXCL, C-X-C motif ligand; FGF1, fibroblast growth factor; HSC, hematopoietic stem cells; SCF, stem cell factor; and TGF, transforming growth factor.

**Table 3 jcm-11-00706-t003:** Definition and prevalence of anemia: alternative definitions and the prevalence of anemia in subsets of elderly individuals.

Definitions of Anemia [95,96,99]		
	Definition Men	Definition Women
**WHO definition of anemia (Hb)**	<13 g/dL	<12 g/dL
**Alternative definitions**	<12 g/dL	<12 g/dL
	Decrease in Hb > 2 g/dL	Decrease in Hb > 2 g/dL
**Prevalence of Hemoglobin Levels in Elderly Individuals** [97]		
**Percent of Individuals**	**Criteria Men**	**Criteria Women**
22.0%	<14 g/dL	<13 g/dL
5.6%	<13.2 g/dL	<12.2 g/dL
3.8%	<13.0 g/dL	<12.0 g/dL
0.6%	<11.0 g/dL	<11 g/dL
**Prevalence of Anemia in Various Subsets of Elderly Patients** [99,100,101]		
**Percent of Individuals**	**Subset of Elderly Individuals**	
12%	Elderly living in private homes	
47%	Elderly living in nursery homes	
40%	Elderly admitted to hospital	

**Table 4 jcm-11-00706-t004:** Causes of anemia in elderly patients, a summary of the results from selected previous studies [98,99,102,103]. For a detailed discussion with additional references see Section 3.5 and Section 3.6.

Cause of Anemia	Percent of Patients
Total fraction: malnutrition, specific deficiencies	20%
Folic acid/cobalamin deficiency	12–15%
Iron deficiency	20%
Renal failure	8–10%
Other chronic diseases, including inflammatory diseases	20%
Renal failure combined with another chronic disease	<5%
Multiple etiologies	20%
Unknown cause of anemia	30–35%

**Table 5 jcm-11-00706-t005:** Suggested initial laboratory evaluation of elderly patients with anemia [106].

Type of Marker	Recommended Single Analyses (Peripheral Blood)
Peripheral blood cells	Hemoglobin, MCV, MCH, differential blood cell count, reticulocyte count, reticulocyte hemoglobin and erythropoietin
Nutritional status	Vitamin B12, serum folate, transferrin saturation and ferritin
Hemolysis	Lactate dehydrogenase, haptoglobin and bilirubin
Organ markers	Creatinine and glomerular filtration rate Alanine aminotransferase and aspartate aminotransferase
Markers of inflammation	C-reactive protein
Others	Serum electrophoresis and thyrotropin-releasing hormone

**Table 6 jcm-11-00706-t006:** Anemia in elderly community-living individuals, a summary of results from representative and important population studies describing the frequency of anemia and mortality in anemic individuals [98,99,116,117,118,119,120].

Study	Population and Methodology	Observation
Schop et al. [98]	The study included 4152 individuals from the general population above 50 years of age (median age: 75). Newly diagnosed anemia.	After an extensive evaluation in general practice the cause was unclear for 20%, one cause was seen for 59% and multiple etiologies for 22%. The most common single etiologies were anemia of chronic disease and iron deficiency. The frequency of patients with renal anemia increased with age.
Patel et al. [116]	The study included 4089 Americans above 65 years of age.	For non-Hispanic white Americans the mortality increased with the degree of anemia, and the anemia threshold for increased mortality corresponded to 0.4 and 0.2 g/100 mL above the WHO definition of anemia (see Table 2). For black Americans the threshold for increased mortality was 0.7 g/100 mL below the WHO definition.
Guralnik et al. [99]	A population-based study including 39,695 individuals, 5252 of them being older than 65.	Anemia prevalence rates increased after 50 years of age. For individuals ≥65 years of age 11.0% of men and 10.2% of women were anemic, and 20% of individuals ≥85 years of age were anemic. Nutrient deficiency was present in one-third, one-third had renal and/or chronic inflammatory anemia and the anemia was unexplained for one-third. Hb levels <11.0 were observed for 1.6% of men and 2.8% of women. Anemia was most frequent in elderly black people (27.8%) and less frequent in Mexican Americans (10.4%) and white non-Hispanics (9.0%).
Penninx et al. [117]	The study included 3607 individuals aged 71 or older, with a mean age of 78.2.	Anemia according to the WHO criteria was observed for 12.5%. The mortality was significantly higher for anemic participants (37.0% vs. 22.1%, *p* < 0.001) and they were hospitalized more frequently and spent more days in hospital. These differences remained significant after excluding persons with prevalent disease.
Shavelle et al. [118]	The study included 7171 community-dwelling individuals (aged ≥ 50), 862 of whom were anemic according to the WHO definition.	Significant negative impact of anemia on overall survival with relative risk 1.8 (*p* < 0.001). Relative risk depended on cause: (i) nutritional (2.34, *p* < 0.0001); (ii) chronic renal disease (1.70, *p* < 0.0001); (iii) chronic inflammation (1.48, *p* < 0.0001); and (iv) unexplained (1.26, *p* < 0.01).
Zakai et al. [119]	The development of anemia was evaluated for 3758 community-dwelling individuals aged 65 or older without anemia at inclusion.	Of the individuals, 498 (8.5%) developed anemia according to the WHO criteria. Baseline increasing age, being African American and kidney disease predicted anemia development over 3 years. Both anemia development and hemoglobin decline predicted subsequent mortality in men and women.
den Enzen [120]	A population-based study of 562 individuals aged 85.	The prevalence of anemia at baseline was 26.7%, and anemic individuals had more comorbidity with more disabilities, worse cognitive function and more depressive symptoms. Both prevalent and incident anemia was significantly associated with survival in adjusted analyses, including adjustment for C-reactive protein. Mortality increased with severity of anemia.

**Table 7 jcm-11-00706-t007:** Anemia in elderly nursing home residents, a summary of results from representative and important population studies [99,121,122,123,124,125,126].

Study	Population and Methodology	Observation
Chan et al. [121]	Retrospective, cross-sectional study at nine Chinese nursing homes (812 residents, mean age of 86 years).	A total of 67% were anemic, and the anemic residents were older and had a higher incidence of renal impairment; no significant associations with other comorbidities were observed.
Resnick et al. [122]	Including 451 residents, mean age of 83.7 years.	Anemia was more common among black than white residents; physical capacity was worse in anemic patients.
Westerlind et al. [100]	Including 390 patients (mean age 85.1 years), follow-up 7 years from baseline including Hb for 220 patients.	Prevalence of anemia at baseline was 52% for men and 32% for women. Two-year mortality was 61% for men with and 29% for men without anemia (*p* = 0.001), but for women no significant difference was observed (49% vs. 43%). Increased mortality in anemic men was independent of age, BNP and eGFR. Among men, anemia correlated with BNP/eGFR/CRP; for women, anemia correlated with several inflammatory markers including CRP. Anemic men were less physically active. Reduction in Hb with more than 0.9 g/100 mL during the first 2 years of follow-up was associated with increased mortality.
Pandya et al. [124]	Including 564 residents, mean age of 81 years.	In this study, 64% of males and 53% of females were anemic. Anemia was significantly associated with being African American, low eGFR, cancer, gastrointestinal bleeding and inflammatory disease,
Landi et al. [125]	Including 372 residents admitted to nursing home, aged 65 years or older.	At enrolment 63.1% of patients were anemic according to the WHO criteria. The death rate of anemic patients (38%) was higher than for nonanemic patients (28%, *p* = 0.03). This difference was independent of frailty, cognitive impairment, eGFR, cancer, stroke, body mass index and pressure ulcer.
Robinson et al. [126]	Evaluated 6200 residents, mean age of 83.2 years.	Of the residents, 59.6% were anemic. Older age was associated with lower hemoglobin in patients without kidney disease. However, for the whole study population chronic kidney disease seemed to contribute more strongly to the development of anemia than high age.

Abbreviations: BNP, B-type natriuretic peptide; CRP, C-reactive protein; and eGFR, estimated glomerular filtration rate.

**Table 8 jcm-11-00706-t008:** Inflammation and cardiovascular disease. A summary of observations from important studies [198,199,200,201,202,203].

Study and Study Population	Observation
Tracy et al. [198]. Prospective, nested case–control study. There were 5201 persons in the original sample, age ≥ 65. Of these, 146 cases with incident cardiovascular events were identified and 146 control subjects were matched (mean age of 72.8 and 72.9 years, respectively).	The mean CRP level was only significantly higher for case subjects than for control subjects in women. CRP levels were generally higher in persons with subclinical disease. Among the elderly with subclinical disease, CRP was associated with myocardial infarction with an overall odds ratio (OR) of 2.67 (CI 1.04–6.81), with the association being stronger for women, with an OR of 4.50 (CI 0.97 to 20.8), than for men, with an OR of 1.75 (CI 0.51 to 5.98).
Cesari et al. [199]. Cross-sectional study including 3045 well-functioning persons with a mean age of 74.2 years (range of 70–79 years). Subclinical cardiovascular disease was defined as signs of angina or claudication according to the Rose Questionnaire, positive ankle/brachial index or electrocardiographic abnormalities.	CRP was significantly associated with congestive heart failure, with an OR of 1.64 (95% CI 1.11 to 2.41), but not with any other manifestations of cardiovascular disease. When comparing patients in the highest versus those in the lowest IL6 tertile, the OR for subclinical cardiovascular disease was 1.58 (95% CI 1.26 to 1.97) and for clinical cardiovascular disease was 2.35 (95% 1.79 to 3.09). This was similar for TNFα, with an OR of 1.48 (95% CI 1.16 to 1.88) and 2.05 (95% CI 1.55 to 2.72), respectively. Only soluble TNFR1 but not soluble IL6R or soluble IL2R showed a significant association with cardiovascular disease.
Cesari et al. [200]. Prospective study including 2225 persons with a mean age of 74.0 years (range of 70–79 years). Outcomes were hospitalizations for coronary heart disease (CHD), stroke or congestive heart failure (CHF).	CRP was significantly associated only with congestive heart failure. In contrast, IL6 was significantly associated with all three outcomes, and TNFα was significantly associated both with coronary heart disease and congestive heart failure. The risk of both coronary heart disease and congestive heart failure was highest for patients with levels in the highest tertile for all three markers; these patients had a two- to three-fold increase in coronary disease and heart failure compared with patients with no levels in the highest tertile. These differences also remained significant in adjusted analyses.
Makita et al. [201]. Cross-sectional study including 2056 individuals with a mean age of 58.3 years (range of 25–86 years). All examined with CRP and carotid ultrasound.	An association between plaque score and increasing CRP levels was seen only for men (*p* < 0.01); this association remained significant after being adjusted for age and other risk factors. Intima–media complex thickness and arterial dilation showed significant associations with CRP only in univariate, but not in adjusted, analyses.
Hosford-Donovan et al. [202]. Cross-sectional study of 108 elderly women with a mean age of 67.5 years (range of 65–70 years).	Body mass index (BMI), waist circumference, systolic blood pressure (SBP) and diastolic blood pressure (DBP) were significantly higher in patients with high CRP above the median level. SBP and DBP remained significantly higher in the high-CRP group after adjusting for BMI and use of antihypertensive medication. The influence of CRP on SBP was attenuated when adjusted for waist circumference (*p* = 0.062). Serum derived from high-CRP patients decreased the proliferation and the capillary tube length of in vitro cultured endothelial cells.
Labonté et al. [203]. Cross-sectional study of 801 Inuits (mean age of 36.3 years, range of 18–74 years).	Increased plasma CRP levels >2.0 mg/L were more prevalent among women. SBP was significantly and independently associated with increased CRP levels.

Abbreviations: CHD, coronary heart disease; CI, confidence interval; DBP, diastolic blood pressure; IL2R, IL2 receptor; IL6R, IL6 receptor; OR, odds ratio; RR, relative risk; SBP, systolic blood pressure; and TNFR1, TNF receptor 1.

## Data Availability

Not applicable.

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
