# Peer review of "Hematopoiesis, Inflammation and Aging—The Biological Background and Clinical Impact of Anemia and Increased C-Reactive Protein Levels on Elderly Individuals"

_jcm, 2022, doi:10.3390/jcm11030706_

Round 1

Reviewer 1 Report

In this manuscript, Bruserud et al. have synthesized an effective discussion on inflammation and aging and a need for personalized interventions for elderly individuals. This is a compelling review for the readers of this journal. Here are my brief comments:

  1. In the introduction, the authors can try to provide numbers to put the importance of this field into perspective. For example, According to WHO “By 2050, the world's population of people aged 60 years and older will double (2.1 billion). The number of persons aged 80 years or older is expected to triple between 2020 and 2050 to reach 426 million.”
  2. Authors can improve figure 2 (provide an enriched figure) by describing individual variables in 1 sentence and adding a reference: For example:
  3. Genomic instability – mutations and chromosomal aberrations within the genome (Reference, PMID: 28550046). Similarly for others.
  4. Above reference: Vijg, J., Dong, X., Milholland, B. and Zhang, L., 2017. Genome instability: a conserved mechanism of ageing? Essays in biochemistry61(3), pp.305-315.
  5. Authors have mentioned “Monocytes/macrophages: These cells can be polarized by CRP towards the proin-826 flammatory M1 phenotype with increased phagocytosis and cytokine release, in-827 hibited chemotaxis and altered metabolism with increased LDL uptake.” A small section can be dedicated macrophage polarization can be discussed here. For example: ‘ging causes a shift in macrophage polarization from anti-inflammatory 'M2' to pro-inflammatory 'M1' that is associated with a rise in cytokines and immune cells in the ENS.’ From reference: Becker, Laren, Linh Nguyen, Jaspreet Gill, Subhash Kulkarni, Pankaj Jay Pasricha, and Aida Habtezion. "Age-dependent shift in macrophage polarisation causes inflammation-mediated degeneration of enteric nervous system." Gut 67, no. 5 (2018): 827-836.
  6. The conclusion can be expanded for better emphasis.

Author Response

We are very grateful for the reviewer’s comments and for the opportunity to submit a Revised Version of our article. We have carefully considered the reviewer’s comments and our response to each comment is given below. All our alterations in the Revised Version are marked with yellow. The new reference numbering is marked with blue in the Revised Version.

Our review is relatively long, and we are especially grateful for the opportunity to present a review that includes a detailed discussion of three large and complex scientific field that are related and partly overlapping/interacting; i.e. aging hematopoiesis, anemia in elderly, inflammation in elderly.

RESPONSE TO REVIEWER 1

In this manuscript, Bruserud et al. have synthesized an effective discussion on inflammation and aging and a need for personalized interventions for elderly individuals. This is a compelling review for the readers of this journal.

Response: We are very grateful for this general comment.

1.1.In the introduction, the authors can try to provide numbers to put the importance of this field into perspective. For example, According to WHO “By 2050, the world's population of people aged 60 years and older will double (2.1 billion). The number of persons aged 80 years or older is expected to triple between 2020 and 2050 to reach 426 million.”

Response: We have added a new chapter to the introduction (page 1), and one new updated reference is added.

1.2. Authors can improve figure 2 (provide an enriched figure) by describing individual variables in 1 sentence and adding a reference: For example: Genomic instability – mutations and chromosomal aberrations within the genome (Reference, PMID: 28550046). Similarly for others.

Above reference: Vijg, J., Dong, X., Milholland, B. and Zhang, L., 2017. Genome instability: a conserved mechanism of ageing? Essays in biochemistry61(3), pp.305-315.

Response: We suppose this is a printing error and that the reviewer means Figure 1 where Genomic instability is mentioned as the first of the nine cellular hallmarks of aging. We have revised the figure as suggested by the reviewer and new references have been added for each of the term, these references are carefully selected recent reviews (references 3-11). We have also used the additional references in corresponding sections in the article. We hope our solutions can be accepted.

1.3.Authors have mentioned “Monocytes/macrophages: These cells can be polarized by CRP towards the proin-826 flammatory M1 phenotype with increased phagocytosis and cytokine release, in-827 hibited chemotaxis and altered metabolism with increased LDL uptake.” A small section can be dedicated macrophage polarization can be discussed here. For example: ‘ging causes a shift in macrophage polarization from anti-inflammatory 'M2' to pro-inflammatory 'M1' that is associated with a rise in cytokines and immune cells in the ENS.’ From reference: Becker, Laren, Linh Nguyen, Jaspreet Gill, Subhash Kulkarni, Pankaj Jay Pasricha, and Aida Habtezion. "Age-dependent shift in macrophage polarisation causes inflammation-mediated degeneration of enteric nervous system." Gut 67, no. 5 (2018): 827-836.

Response: We have now included an additional comment in Section 2.3, the monocyte chapter. The suggested reference is also included (reference 11).

1.4 The conclusion can be expanded for better emphasis.

Response: The conclusion has been rewritten and extended.

Reviewer 2 Report

The authors include the most important papers of the literature. However, since the article resembles more with a book chapter authors could omit in the article information that is included in figures or vice versa in order  the article to be more easy to read.

Author Response

REVIEWER 2

The authors include the most important papers of the literature. However, since the article resembles more with a book chapter authors could omit in the article information that is included in figures or vice versa in order the article to be more easy to read.

Response: We have tried to address this comment in our article. However, our review is quite long (26 pages without references, 225 references) and the molecular and cellular mechanisms behind anemia and inflammation are quite complex. For these reasons we think that summarizing tables can be a help for the reader. However, for the summarizing tables we have now clearly stated in the text and/or the table heading when they are summarizing so that this should be clear for the reader. Our comments with regard to the individual tables are given below.

We would point out that this comment is relevant especially for Figures 1 and 2 and for Tables 1 and 2. This is explained below for each individual table/figure. For the revised Figure 1 we have avoided overlap as suggested by the reviewer. For Figure 2 and for Tables 1 and 2 we clearly state that they represent summaries of the most important information reviewed and discussed in the corresponding text.

We hope that our article can be of interest both for biologists and clinicians. Our intention with the summarizing tables was therefore to try to help the basic biologists to understand the clinical implications of aging-associated cellular alteration, and to help the clinicians to better understand the complex mechanisms behind the common clinical problems anemia and inflammation in elderly. We agree that certain tables show similarities with a textbook presentation, but we think this is useful because the text of our review is longer than many other reviews. In our opinion the textbook-like presentation is helpful for the reader in such a long review (26 pages without references).

It is now clearly stated in the heading of all relevant Tables and in the figure text when these parts represent summaries of the more detailed presentation in the text. We give comments to individual tables and figures below; they are then listed in the same order as they appear in the text.

Figure 1: Reviewer 1 asked us to review this table and include additional information. Based on the general comment given by reviewer 2 we have added more detailed information only in the table but not to the text.

Table 1: It is now clearly stated in the legend that this is a summary of the text in section 2.2.

Figure 2 and Table 2: This is a summary of the hematopoiesis-supportive effects of various stromal cells. This information is scattered throughout the text in Section 2.3; we think it is a help for the reader (especially clinicians) to have these two tables as easily available possibilities to go back and repeat when reading the later parts of the review.

Table 3: Only parts of this information is given in the text and then as parts of more general comments.

Table 4: This Table is a summary of the results from several studies, whereas our discussion in the text is mainly based on the observations in one large and representative study (reference 102).

Table 5: This detailed information is not given in the text.

Tables 6 and 7. Both these tables give more and detailed information from important representative studies. The corresponding text is a general discussion based on the information given in these tables. The clinical consequences associated with anemia/inflammation (i.e. decreased survival) are very important to present and discuss, and we think that it is important to give the reader some additional and more detailed information to illustrate the quality of the studies that form the scientific basis for these important clinical conclusions/consequences. Thus, for these two tables there is no clear overlap with the text.